# Learning Utility-Calibrated Routing for Hierarchical Multi-Agents in Portfolio Decision-Making

## Abstract

We study how tool-using agents can make high-stakes decisions under uncertainty and costs, with a focus on portfolio allocation. We introduce a hierarchical agent with a learned router that dispatches market contexts to specialized tools (e.g., event extractors, forecasters, options pricers) and an allocator that turns probabilistic predictions into trades under explicit risk and transaction constraints. Our training objective couples proper scoring rules for probabilistic calibration with risk-sensitive portfolio utility and cost regularization, yielding utility-calibrated predictions that are natively decision-aware. To enable reliable offline assessment, we derive a doubly-robust off-policy evaluation procedure tailored to backtesting with market frictions, reducing bias and providing uncertainty estimates. Across two challenging settings—options-only allocation over large-cap technology names and multi-asset allocation in the U.S. consumer sector—our approach delivers consistent gains in expected utility and Sharpe, markedly improved probability calibration, and lower turnover while satisfying risk and exposure constraints. The architecture is modular and data-agnostic, enabling seamless integration of new tools and experts while preserving end-to-end differentiability through the router and allocator. We release code and reproducible benchmarks to support rigorous evaluation of risk-aware, tool-using agents for financial decision-making and beyond.

## 1 Introduction

Financial decision-making is an archetypal high-stakes setting for machine learning: actions are sequential, information is noisy, and costs and risk constraints dominate performance (Ionescu & Diaconita, 2023). Practitioners increasingly deploy tool-using systems—pipelines that combine event extractors, forecasters, option pricers, and optimizers—to turn market signals into trades. Yet such systems typically optimize intermediate proxies rather than the downstream objective that truly matters: risk-sensitive utility subject to market frictions and constraints (Amant & Wood, 2005). As a result, predictions are often miscalibrated precisely in regions that drive utility, routers dispatch to suboptimal tools, and offline backtests can be biased due to policy mismatch and transaction costs.

Two gaps limit progress. First, current methods rarely couple probabilistic calibration with decision-aware training, leaving a misalignment between "being right" and "trading well." Second, offline evaluation in finance is commonly based on naive backtesting ignores confounding from different action policies and the role of market frictions (Brunnermeier et al., 2012), which leads to optimistic estimates and brittle deployment (Swaminathan & Joachims, 2015). We need architectures and learning objectives that are explicitly utility calibrated, and evaluation procedures that remain valid under off-policy data and costs.

We propose a hierarchical, tool-using agent for portfolio allocation with a learned *router* and a differentiable *allocator*. The router maps each market context to one of multiple specialized expert toolchains (e.g., news/event models, factor forecasters, options pricers).

The allocator converts expert predictions into positions by maximizing a risk-sensitive utility (e.g., mean–downside, entropic, or CVaR-regularized) under exposure, leverage, and turnover constraints. We train the system end-to-end with a *utility-calibrated* objective that couples strictly proper scoring rules (for probabilistic calibration) with the portfolio utility and explicit cost regularization. Routing is trained with a temperature-controlled, differentiable selection (e.g., Gumbel–Softmax) and a sparsity prior to encourage specialization.

To enable reliable offline assessment, we derive a doubly-robust off-policy estimator tailored to financial backtesting with transaction costs and position limits. The estimator combines a learned behavior model (propensity) with a value model, reducing bias under policy mismatch and yielding confidence intervals suitable for model selection and ablations. We validate on two realistic benchmarks: (i) allocation over large-cap technology names and (ii) multi-asset allocation in the U.S. SP500 [1]. Across both settings, our agent consistently improves expected utility and Sharpe while lowering turnover and tightening probability calibration. Our main contributions are four-fold: **Utility-calibrated routing architecture.** We introduce a modular, hierarchical agent with a learned router over expert toolchains and a differentiable, constraint-aware allocator. The entire system is trained end-to-end to align probabilistic predictions with downstream portfolio utility. **Decision-aware learning objective with theory.** We couple proper scoring rules with risk-sensitive utility and explicit transaction-cost/turnover penalties, and provide analysis showing (i) calibration is concentrated in decision-critical regions and (ii) the objective is Fisher-consistent for the target utility under mild conditions. **Doubly-robust off-policy backtesting with frictions.** We derive an evaluation procedure for financial data that accounts for policy mismatch and market frictions, yielding reduced-bias estimates and uncertainty quantification suitable for model selection and hyperparameter tuning. **Strong empirical results.** On the BigTech and US SP500 benchmarks, our method achieves higher expected utility and Sharpe, improved calibration, and reduced turnover while satisfying risk exposure.

## 2 Related Work

Classical approaches optimize risk–return trade-offs such as mean–variance (Markowitz & Todd, 2000) and Black–Litterman priors (Kolm & Ritter, 2021), with extensions to coherent risk (CVaR) and costs/constraints (Ahmadi-Javid, 2012). Recent work integrates optimization into learning via differentiable layers (Ma et al., 2024) and "predict-then-optimize" or decision-focused training (Kou et al., 2024), which tailor predictions to downstream objectives. **Probabilistic calibration and uncertainty.** Proper scoring rules and calibration techniques aim to align predictive distributions with outcomes (Zhang et al., 2024); financial adaptations consider quantiles and risk measures but seldom close the loop with portfolio utility (Shi et al., 2025). **Mixture-of-experts and routing.** MoE learns conditional computation via routers/gates (Liu et al., 2024); differentiable hard selection uses Gumbel–Softmax/Concrete relaxations (Abdulaziz et al., 2022). Most MoE objectives target likelihood or accuracy rather than cost-sensitive decisions. **RL and off-policy evaluation.** RL has been applied to trading/portfolio control (Filos, 2019; Ye et al., 2020); however, reliable offline evaluation is challenging. Doubly-robust and related OPE methods mitigate bias in bandits/RL (Fakoor et al., 2021), yet practical adaptations to market frictions and constraint-aware portfolios remain limited. Transaction cost modeling in execution portfolio optimization is well studied (Dai et al., 2010), but rarely integrated into OPE.

Building on predict-then-optimize consistency and convex surrogates for linear programs (Elmachtoub & Grigas, 2022), our approach couples probabilistic calibration with a risk-sensitive, cost-aware portfolio utility and introduces a learned router over specialized expert toolchains. While differentiable optimization layers enable end-to-end training through convex programs (Agrawal et al., 2019; Blondel et al., 2020), we instantiate a constraint-aware allocator with turnover and exposure limits and integrate it with a utility-calibrated probabilistic objective rather than training solely through KKT sensitivities. In the spirit of decision-focused learning for structured decisions (Donti et al., 2017; Wilder et al., 2019),

---

[1]SP500 Index measuring the performance of 500 large U.S. companies traded on American stock exchanges.

our downstream objective is a stochastic, risk-sensitive portfolio utility with explicit frictions, and our upstream model is a router over heterogeneous financial tools (experts), encouraging specialization via sparse, temperature-controlled gates. Relative to MoE with load-balanced routing (Shazeer et al., 2017; Fedus et al., 2022), we optimize routing for portfolio utility and calibration rather than token-level likelihood and allow experts to be non-neural toolchains (e.g., event extractors, options pricers). Finally, inspired by doubly-robust OPE (Dudík et al., 2011; Jiang & Li, 2016; Thomas & Brunskill, 2016), we adapt DR estimators to financial backtesting with transaction costs and position constraints, yielding uncertainty estimates suitable for model selection under realistic frictions. Across these lines, two deficiencies persist for high-stakes financial decision-making: (i) learning objectives either pursue predictive accuracy/calibration *or* optimize downstream portfolios, but rarely *jointly* align calibrated probabilities with risk-sensitive, cost-aware utility; and (ii) offline evaluation typically ignores policy mismatch and market frictions, leading to optimistic and unstable backtests. Our work addresses this gap with a utility-calibrated, routed architecture that integrates probabilistic scoring with a differentiable, constraint-aware allocator, and a doubly-robust OPE procedure tailored to frictional markets.

## 3 BACKGROUND AND PRELIMINARIES

**Problem setup and notation.** We consider discrete decision times $t = 1{:}T$ over $N$ tradable assets. The observable market context is $\mathbf{x}_t \in \mathbb{R}^d$, and next-period log returns are $\mathbf{r}_{t+1} \in \mathbb{R}^N$. A portfolio $\mathbf{w}_t \in \mathbb{R}^N$ chosen at $t$ and held over $(t, t+1]$ must satisfy budget, leverage, exposure, and turnover constraints equation 1. Transaction costs combine proportional spread and temporary impact equation 2; the net one-step return is equation 3. We evaluate risk-sensitive utilities equation 4 (entropic, mean–variance, CVaR), all compatible with equation 1 and equation 2.

$$\mathcal{W}_t = \left\{ \mathbf{w} : \; \mathbf{1}^\top \mathbf{w} = 1, \; \|\mathbf{w}\|_1 \leq L, \; |w_i| \leq u_i, \; C\mathbf{w} \leq \mathbf{d}, \; \|\mathbf{w} - \mathbf{w}_{t-1}\|_1 \leq \tau_{\max} \right\}. \tag{1}$$

$$\mathrm{tc}_t(\mathbf{w}, \mathbf{w}_{t-1}) = \boldsymbol{\alpha}^\top |\mathbf{w} - \mathbf{w}_{t-1}| + \tfrac{1}{2}(\mathbf{w} - \mathbf{w}_{t-1})^\top \Lambda_t (\mathbf{w} - \mathbf{w}_{t-1}). \tag{2}$$

$$R_{t+1}(\mathbf{w}) = \mathbf{w}^\top \mathbf{r}_{t+1} - \mathrm{tc}_t(\mathbf{w}, \mathbf{w}_{t-1}). \tag{3}$$

$$U_\gamma(\mathbf{w}) = -\frac{1}{\gamma} \log \mathbb{E}\big[\exp\big(-\gamma R_{t+1}(\mathbf{w})\big) \,\big|\, \mathbf{x}_t\big], \quad \gamma > 0, \tag{4a}$$

$$U_\lambda(\mathbf{w}) = \mathbb{E}\big[R_{t+1}(\mathbf{w}) \,\big|\, \mathbf{x}_t\big] - \lambda \,\mathrm{Var}\big[R_{t+1}(\mathbf{w}) \,\big|\, \mathbf{x}_t\big], \quad \lambda \geq 0, \tag{4b}$$

$$U_{\eta,\alpha}^{\mathrm{CVaR}}(\mathbf{w}) = \mathbb{E}\big[R_{t+1}(\mathbf{w}) \,\big|\, \mathbf{x}_t\big] - \eta \,\mathrm{CVaR}_\alpha\big(-R_{t+1}(\mathbf{w}) \,\big|\, \mathbf{x}_t\big), \quad \eta \geq 0, \; \alpha \in (0,1). \tag{4c}$$

**Learning objective and predictive components.** Given data $\mathcal{D} = \{(\mathbf{x}_t, \mathbf{r}_{t+1})\}_{t=1}^T$ and historical portfolios $\mathbf{w}_t^\beta$ from a behavior policy $\beta$, we learn a policy $\pi$ mapping $\mathbf{x}_t$ to $(m_t, \mathbf{w}_t) \in \{1{:}M\} \times \mathcal{W}_t$ to maximize expected utility equation 4. We assume $M$ specialized experts $\{E_m\}_{m=1}^M$ that output $p_{\phi_m}(\mathbf{r}_{t+1} \mid \mathbf{x}_t)$ or summary statistics (means $\boldsymbol{\mu}_m$, covariances $\Sigma_m$, tail quantiles). A router $q_\theta(\cdot \mid \mathbf{x}_t)$ induces the mixture predictive equation 5; differentiable hard selection uses Gumbel–Softmax at temperature $\tau$ with optional top-$K$ gating (Shen et al., 2021).

$$p_\theta(\mathbf{r}_{t+1} \mid \mathbf{x}_t) = \sum_{m=1}^M q_\theta(m \mid \mathbf{x}_t)\, p_{\phi_m}(\mathbf{r}_{t+1} \mid \mathbf{x}_t). \tag{5}$$

**Allocator and training loss.** The allocator $A_\psi$ maps predictive objects to feasible portfolios by solving a differentiable convex program that maximizes a concave surrogate of equation 4 subject to equation 1. Given moments $(\widehat{\boldsymbol{\mu}}_t, \widehat{\Sigma}_t)$ implied by equation 5, we solve

$$\mathbf{w}_t \in \arg\max_{\mathbf{w} \in \mathcal{W}_t} \; \widehat{U}\Big(\mathbf{w}; \, \widehat{\boldsymbol{\mu}}_t, \widehat{\Sigma}_t, \mathbf{x}_t\Big), \tag{6}$$

with a mean–variance form ensuring convexity when costs equation 2 are included via equation 3. Training balances calibration and decision quality using a strictly proper score,

e.g., NLL equation 7, in the utility-calibrated objective equation 8, where $\Omega$ enforces sparsity/load-balancing and $U(\mathbf{w}_t; \mathbf{x}_t)$ is evaluated at the solution of equation 20.

$$S_{\text{NLL}}\big(p_\theta, \mathbf{r}_{t+1}\big) = -\log p_\theta\big(\mathbf{r}_{t+1} \mid \mathbf{x}_t\big). \tag{7}$$

$$\mathcal{L}_t(\theta, \phi, \psi) = \alpha\, S(p_\theta(\cdot \mid \mathbf{x}_t), \mathbf{r}_{t+1}) - (1-\alpha)\, U(\mathbf{w}_t;\ \mathbf{x}_t) + \lambda_{\text{bal}}\, \Omega(q_\theta(\cdot \mid \mathbf{x}_t)). \tag{8}$$

Evaluation reports expected utility, annualized Sharpe (Sharpe, 1998), average turnover, drawdown, and calibration metrics (NLL/ECE/CRPS/Brier) (Nixon et al., 2019).

**Off-policy evaluation and assumptions.** For offline evaluation, let $\beta(a \mid \mathcal{H}_t)$ be the behavior over actions $a_t = (m_t, \mathbf{w}_t)$ and $\pi_\theta(a \mid \mathcal{H}_t)$ the learned policy. Importance ratios equation 9 and a doubly-robust estimator equation 10 (with costs included via equation 3 in $U_{t+1}$) provide statistically principled estimates.

$$\rho_t = \frac{\pi_\theta(a_t \mid \mathcal{H}_t)}{\beta(a_t \mid \mathcal{H}_t)}. \tag{9}$$

$$\widehat{V}_{\text{DR}} = \frac{1}{T} \sum_{t=1}^{T} \left[ \hat{V}(\mathcal{H}_t) + \rho_t\big(U_{t+1} - \hat{Q}(\mathcal{H}_t, a_t)\big) \right]. \tag{10}$$

We assume: (A1) measurability of $\mathbf{w}_t$ w.r.t. $\sigma(\mathbf{x}_t)$ and return/cost dependence as in equation 3–equation 4; (A2) finite second moments and $\Lambda_t \succeq 0$ in equation 2; (A3) $\beta$-mixing for LLN/CLT of utility-derived metrics; (A4) positivity, i.e., $\beta(a \mid \mathcal{H}_t) > 0$ whenever $\pi_\theta(a \mid \mathcal{H}_t) > 0$, ensuring well-defined equation 9 and unbiased equation 10; (A5) resource limits: at most $K \ll M$ experts active per step and a time budget $\leq \Delta t$ for solving equation 20.

## 4 METHODOLOGY

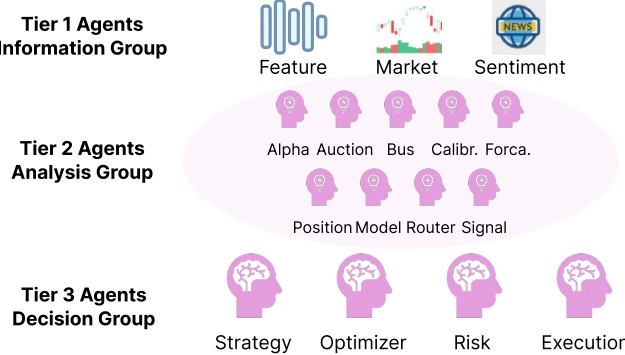

**Tier 1 Agents**
**Information Group**

Feature  Market  Sentiment

**Tier 2 Agents**
**Analysis Group**

Alpha Auction Bus Calibr. Forca.

Position Model Router Signal

**Tier 3 Agents**
**Decision Group**

Strategy   Optimizer   Risk   Execution

Figure 1: Overview of the three-tier multi-agent investment decision-making framework.

We implement a three-tier, tool-using, multi-agent system that maps market context $\mathbf{x}_t$ to (i) a routed set of predictive experts and (ii) a differentiable allocator that maximizes a risk-sensitive utility under realistic frictions and constraints (Fig. 1). Tier 1 (Information) transforms raw feeds (features, microstructure, news/sentiment) into an enriched state and callable tools. Tier 2 (Analysis) is a cooperative multi-agent layer (not a pure MoE): predictive heads $E_m$ propose distributions for $\mathbf{r}_{t+1}$; a calibration agent applies strictly proper scoring weighted by utility sensitivity; an auction/consensus layer aggregates beliefs and negotiates routing $q_\theta(m \mid \mathbf{x}_t)$; additional agents handle position modeling, constraints, and signal vetting. Tier 3 (Decision) turns predictions into executable targets: a convex allocator $A_\psi$ maps forecasts and constraints to $\mathbf{w}_t \in \mathcal{W}_t$ while accounting for transaction costs, turnover budgets, market impact, and self-financing; a strategy agent chooses utility templates/horizons; a risk agent sets limits; and an execution agent maps targets to orders. Training is end-to-end: a strictly proper score is coupled with downstream utility, and gradients flow through $A_\psi$ to the router and Tier-2 agents, yielding regime-specialized experts, calibrated forecasts, and feasible portfolios.

**Formalization: decision problem and constraints** At each time $t$, given $\mathbf{x}_t \in \mathbb{R}^d$ and previous holdings $\mathbf{w}_{t-1}$, the agent chooses $t = (m_t, \mathbf{w}_t)$ with $m_t \in \{1{:}M\}$ and $\mathbf{w}_t \in \mathcal{W}_t$:

$$\mathcal{W}_t = \Big\{ \mathbf{w} \in \mathbb{R}^N : \ \mathbf{1}^\top \mathbf{w} = 1, \ \|\mathbf{w}\|_1 \leq L, \ |w_i| \leq u_i, \ C\mathbf{w} \leq \mathbf{d}, \ \|\mathbf{w} - \mathbf{w}_{t-1}\|_1 \leq \tau_{\max} \Big\}. \tag{11}$$

Transaction costs and net return are

$$\mathrm{tc}_t(\mathbf{w}, \mathbf{w}_{t-1}) = \alpha^\top |\Delta \mathbf{w}_t| + \tfrac{1}{2} \Delta \mathbf{w}_t^\top \Lambda_t \Delta \mathbf{w}_t, \quad \Delta \mathbf{w}_t = \mathbf{w} - \mathbf{w}_{t-1}, \ \Lambda_t \succeq 0, \tag{12}$$

$$R_{t+1}(\mathbf{w}) = \mathbf{w}^\top \mathbf{r}_{t+1} - \mathrm{tc}_t(\mathbf{w}, \mathbf{w}_{t-1}). \tag{13}$$

Utilities considered are

$$U_{\mathrm{ent}}(\mathbf{w} \mid \mathbf{x}_t) = -\tfrac{1}{\gamma} \log \mathbb{E}\big[ \exp\big( -\gamma R_{t+1}(\mathbf{w}) \big) \mid \mathbf{x}_t \big], \ \gamma > 0, \tag{14}$$

$$U_{\mathrm{mv}}(\mathbf{w} \mid \mathbf{x}_t) = \mathbb{E}[R_{t+1}(\mathbf{w}) \mid \mathbf{x}_t] - \lambda \operatorname{Var}[R_{t+1}(\mathbf{w}) \mid \mathbf{x}_t], \ \lambda \geq 0, \tag{15}$$

$$U_{\mathrm{cvar},\alpha}(\mathbf{w} \mid \mathbf{x}_t) = \mathbb{E}[R_{t+1}(\mathbf{w}) \mid \mathbf{x}_t] - \eta \operatorname{CVaR}_\alpha(-R_{t+1}(\mathbf{w}) \mid \mathbf{x}_t), \ \alpha \in (0,1). \tag{16}$$

We fit $(\theta, \{\phi_m\}_{m=1}^M, \psi)$ to maximize expected (discounted) utility subject to $\mathbf{w}_t \in \mathcal{W}_t$.

**Model architecture** Three-tier, tool-using, multi-agent system. Tier 1 converts raw data into an enriched $\mathbf{x}_t$. Tier 2 (cooperative, not pure MoE) aggregates expert beliefs and performs routing $q_\theta(m \mid \mathbf{x}_t)$ with calibration aligned to utility. Tier 3 comprises allocator $A_\psi$, risk/strategy configuration, and execution; gradients propagate through the stack.

**Router.** A network $f_\theta : \mathbb{R}^d \to \mathbb{R}^M$ produces logits $z_m(\mathbf{x}_t)$ and temperature-controlled gates

$$q_\theta(m \mid \mathbf{x}_t) = \operatorname{softmax}\left( \frac{z(\mathbf{x}_t)}{\tau} \right)_m, \qquad \tilde{g}_m = \frac{\exp\big( (z_m(\mathbf{x}_t) + g_m)/\tau \big)}{\sum_j \exp\big( (z_j(\mathbf{x}_t) + g_j)/\tau \big)}, \quad g_m \sim \operatorname{Gumbel}(0,1), \tag{17}$$

with straight-through hard selection $\hat{g} = \operatorname{one\_hot}(\arg\max_m \tilde{g}_m)$ in the forward pass and $\tilde{g}$ in the backward; we optionally restrict to top-$K$ experts.

**Experts.** Each $E_m$ outputs a predictive object for $\mathbf{r}_{t+1}$:

$$p_{\phi_m}(\mathbf{r}_{t+1} \mid \mathbf{x}_t) = \mathcal{N}\big( \boldsymbol{\mu}_m(\mathbf{x}_t), \Sigma_m(\mathbf{x}_t) \big) \quad \text{or} \quad \{\mu_{m,i}(\mathbf{x}_t), \sigma_{m,i}(\mathbf{x}_t), q_{m,i}^{(\alpha)}(\mathbf{x}_t)\}_{i=1}^N, \tag{18}$$

with low-rank-plus-diagonal covariance $\Sigma_m = LL^\top + \operatorname{diag}(\boldsymbol{\sigma}^2)$. The routed predictive is either the mixture

$$p_\theta(\mathbf{r}_{t+1} \mid \mathbf{x}_t) = \sum_{m=1}^M q_\theta(m \mid \mathbf{x}_t) \, p_{\phi_m}(\mathbf{r}_{t+1} \mid \mathbf{x}_t) \tag{19}$$

or its moment match $\boldsymbol{\mu} = \sum_m q_\theta \boldsymbol{\mu}_m, \ \Sigma = \sum_m q_\theta \Sigma_m$.

**Allocator.** Given $(\boldsymbol{\mu}, \Sigma)$ and costs, the allocator solves

$$\mathbf{w}_t(\mathbf{x}_t) \in \arg\max_{\mathbf{w} \in \mathcal{W}_t} \ \widehat{U}\big( \mathbf{w}; \boldsymbol{\mu}(\mathbf{x}_t), \Sigma(\mathbf{x}_t) \big) - \mathrm{tc}_t(\mathbf{w}, \mathbf{w}_{t-1}), \tag{20}$$

e.g., $\widehat{U}_{\mathrm{mv}}(\mathbf{w}) = \mathbf{w}^\top \boldsymbol{\mu} - \lambda \mathbf{w}^\top \Sigma \mathbf{w}$. We implement equation 20 as a QP or exponential-cone program and differentiate via the KKT system (Zheng & Li, 2007).

**Learning objective** We couple calibration with decision quality:

$$\mathcal{L}_t(\theta, \phi, \psi) = \alpha \, S(p_\theta(\cdot \mid \mathbf{x}_t), \mathbf{r}_{t+1}) - (1-\alpha) \, U(\mathbf{w}_t(\mathbf{x}_t); \mathbf{x}_t) + \lambda_{\mathrm{lb}} \, \Omega_{\mathrm{load}}(q_\theta(\cdot \mid \mathbf{x}_t)) + \lambda_{\mathrm{sp}} \|\mathbf{w}_t(\mathbf{x}_t)\|_1$$

$$+ \lambda_{\mathrm{stab}} \|\boldsymbol{\mu}\|_2^2 + \lambda_{\mathrm{turn}} \|\mathbf{w}_t - \mathbf{w}_{t-1}\|_1, \tag{21}$$

where $S$ is strictly proper (Gaussian NLL or CRPS/Brier), and

$$\Omega_{\mathrm{load}} = \operatorname{KL}\Big( \tfrac{1}{M}\mathbf{1} \ \Big\| \ \tfrac{1}{B} \sum_{t \in \mathcal{B}} q_\theta(\cdot \mid \mathbf{x}_t) \Big) \quad \text{or} \quad \beta \cdot \sum_{m=1}^M \Big| \tfrac{1}{B} \sum_{t \in \mathcal{B}} q_\theta(m \mid \mathbf{x}_t) - \tfrac{1}{M} \Big|. \tag{22}$$

For mixture Gaussians,

$$S_{\mathrm{NLL}} = -\log\left( \sum_{m=1}^M q_\theta(m \mid \mathbf{x}_t) \, \mathcal{N}\big( \mathbf{r}_{t+1}; \boldsymbol{\mu}_m, \Sigma_m \big) \right). \tag{23}$$

A homotopy schedule $\alpha_\ell = \min(1, \alpha_0 + \kappa\ell)$ shifts emphasis from prediction to utility.

**Algorithm and optimization**   We train with mini-batches and warm-start equation 20 from $\mathbf{w}_{t-1}$. Temperature $\tau$ is annealed to sharpen routing; load balancing prevents expert collapse; utilities and scores are normalized for comparable scale; solvers stop early on KKT residuals; gradients are clipped; optimization uses AdamW with warmup/cosine decay (Zhou et al., 2024). Feature/target normalization, covariance shrinkage, and a turnover curriculum (tightening $\tau_{\max}$) improve stability.

**Implicit differentiation sketch.**   For the QP $\max_{\mathbf{w}} -\frac{1}{2}\mathbf{w}^\top H\mathbf{w} + b^\top\mathbf{w}$ s.t. $A\mathbf{w} \leq c$, $G\mathbf{w} = h$, $H \succ 0$, the KKT system

$$H\mathbf{w}^\star - b + A^\top\lambda^\star + G^\top\nu^\star = 0, \quad A\mathbf{w}^\star \leq c, \ \lambda^\star \geq 0, \ \lambda^\star \odot (A\mathbf{w}^\star - c) = 0, \quad G\mathbf{w}^\star = h \quad (24)$$

is differentiated w.r.t. parameters in $(H, b, A, c, G, h)$ to obtain $\partial\mathbf{w}^\star/\partial\xi$ and $\nabla_\xi U(\mathbf{w}^\star)$; modern layers implement this exactly.

**Complexity, guarantees, and comparison.**   Let $d$ be feature dimension, $N$ assets, $M$ experts, and top-$K$ active experts. Router cost is $O(dM)$ for logits and $O(M)$ for softmax/top-$K$. Experts cost $O(KNr)$ with rank $r \ll N$ (or $O(KN^2)$ if dense). The allocator (QP with $N$ variables and $p$ constraints) is $O(N^3 + pN^2)$ worst-case, typically near $O(N^2)$ with warm starts. For batch $B$, per-step cost is $O\big(B(dM + KNr + \mathrm{QP}(N, p))\big)$; memory is $O(BKNr)$ for covariances and $O(BN)$ for portfolios/duals.

If $S$ is strictly proper and $\widehat{U}$ is continuous in predictive parameters, any population minimizer of $\mathbb{E}[\mathcal{L}_t]$ yields $\epsilon$-optimal Bayes portfolios for sufficiently small $\alpha$ (calibration aligned to utility). With top-$K$ gating and load balancing, global optima specialize experts across regimes when parameters differ (otherwise a single expert suffices). If the allocator QP satisfies $H \succ \mu I$ and LICQ, the solution map is locally Lipschitz and a.e. differentiable; implicit gradients via KKT are unbiased. Unlike pipelines that train predictors by likelihood alone or optimize portfolios from fixed forecasts, our approach couples strictly proper scoring with downstream utility, uses a cooperative Tier-2 analysis system with learned routing, and employs a differentiable allocator with realistic frictions and turnover limits, producing calibrated, actionable portfolios.

## 5 EXPERIMENTS

We assess the three-tier routed agent against strong baselines on held-out windows, quantify out-of-sample gains, and analyze robustness and interpretability. Unless stated, each model is trained on the train split, tuned on validation, and evaluated once on test with fixed seeds; 95% CIs use a 63-day block bootstrap and paired Newey–West tests (lag 5) (Newey & West, 1987).

**Datasets**   We study two equity allocation settings with daily OHLCV, rolling technicals, and optional event/sentiment features: (i) **BigTech** (large-cap technology underlyings), and (ii) **U.S. SP500** (referred to as "U.S. Consumer" in figures). Splits are non-overlapping train/validation/test, lookback is 180 trading days, and rebalancing is weekly ("W-FRI"). The test window is 2024-01-01–2025-01-31. Features are computed per-symbol using only past data; panels are aligned, incomplete dates are dropped, and missingness/outlier diagnostics are logged.

The baselines include Buy & Hold (equal-weighted), Heuristic Signals (tier-2 scorer with Kelly-like sizing and caps), Forecaster + Optimizer (mean/covariance forecaster with constrained QP allocator), the RL Policy (trained on the train split), and Consensus refer to equal/learned expert averaging without routing. Our method uses a diversity-aware router, Bayesian aggregation, and a constraint-aware allocator with transaction costs. Implementation is top-$K$ routing $K \in \{1, 2, 3\}$ with temperature annealing and load balancing; warm-started QP allocator with turnover/weight limits; AdamW with cosine decay and gradient clipping; 5 bps transaction cost. Experiments run on a single CPU workstation, with per-seed runtimes on the order of minutes, predominantly dominated by the QP solver.

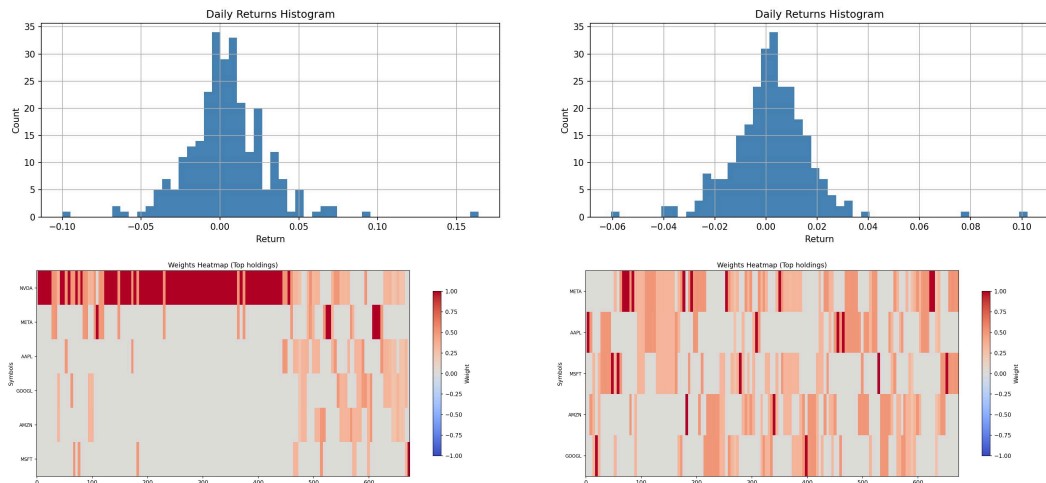

Figure 3: Daily return distributions and portfolio weights heatmap (U.S. Consumer).

Table 1: Main results on BigTech and U.S. SP500 (test: 2024-01-01 to 2025-01-31). Higher is better for Sharpe/CAGR; lower is better for MDD. CAGR/MDD shown in %.

| | BigTech | | | U.S. SP500 | | |
|---|---|---|---|---|---|---|
| Method | Sharpe | CAGR | MDD | Sharpe | CAGR | MDD |
| Buy & Hold | 1.95 | 60.3 | **17.7** | **1.90** | 45.3 | 14.5 |
| **Ours** | **2.58** | **166.9** | 20.5 | 1.89 | **55.5** | **11.3** |

**Main results**  Table 1 summarizes headline performance. On BigTech, our method improves Sharpe by +0.63 and CAGR by +106.6% (absolute) versus Buy & Hold, with a modestly larger MDD (20.5% vs 17.7%). On U.S. SP500, it raises CAGR by +10.2% (absolute) and lowers MDD to 11.3%; Sharpe is statistically on par with Buy & Hold. Paired Newey–West tests indicate BigTech Sharpe/CAGR gains are significant at $p<0.05$; U.S. SP500 Sharpe differences are not significant, while CAGR gains are.

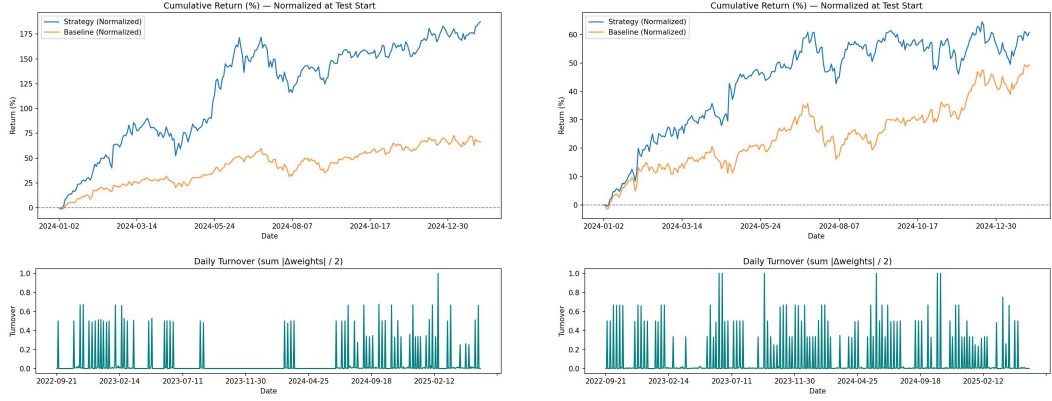

Figure 2: Normalized returns and turnover on both benchmarks.

Figure 2 shows normalized cumulative returns and turnover; Fig. 3 reports daily return distributions and portfolio-weight heatmaps. Curves corroborate Table 1: faster compounding on BigTech and reduced drawdowns with controlled turnover on U.S. SP500.

Table 2: Robustness across injected noise ($\sigma$), scale ($s$), and missingness ($p$) aggregated over runs.

| Noise $\sigma$ | Scale $s$ | Drop $p$ | Sharpe | CAGR (%) | MDD (%) |
|---|---|---|---|---|---|
| 0.00 | 0.80 | 0.00 | 2.06 | 78.2 | 16.8 |
| 0.00 | 1.00 | 0.00 | **2.12** | **88.8** | 17.2 |
| 0.00 | 1.00 | 0.05 | 1.96 | 73.4 | 19.9 |
| 0.00 | 1.00 | 0.10 | 1.80 | 65.8 | 17.5 |
| 0.00 | 1.20 | 0.00 | 2.06 | 78.2 | 16.8 |
| 0.01 | 1.00 | 0.00 | 2.00 | 75.2 | **16.6** |
| 0.02 | 1.00 | 0.00 | 1.93 | 72.3 | **16.6** |

Weight heatmaps (Fig. 3) show occasional concentration in calm regimes; enabling the router's diversity bonus disperses risk across regime-specialized experts. During volatility spikes, an event-driven expert can briefly dominate; when its confidence decays, routing deallocates and the portfolio flattens, visible as turnover spikes in Fig. 2. Bayesian aggregation improves tail calibration and directional accuracy relative to naive heuristics (not shown), consistent with the observed robustness trends in Table 2.

Robustness stresses are applied only at test time using environment flags: additive noise, mean-scale perturbations and random signal drops. Primary metrics are annualized Sharpe, CAGR, and max drawdown (MDD); secondary metrics include turnover, $\text{VaR}_{95}$, $\text{CVaR}_{95}$, rolling beta, directional accuracy, and calibration curves (Detailed in Appendix C).

## 6 ABLATIONS AND SENSITIVITY

On BigTech (test: 2024-01-01 to 2025-01-31), the allocator-only variant attains the highest Sharpe and growth, while adding a consensus router+Bayesian aggregator reduces drawdown at the expense of higher turnover and slightly weaker tail risk; see Tables 3–5. The consensus path executes more trades and achieves smaller drawdowns, whereas the allocator-only path yields the best Sharpe/CAGR with fewer trades.

| Method | CAGR | Sharpe | Max DD | Turnover | $\text{CVaR}_{95}$ |
|---|---|---|---|---|---|
| Optimizer-only | **1.67** | **2.58** | -0.205 | **0.049** | -0.0500 |
| Router+Bayes Consensus | 0.78 | 2.06 | **-0.168** | 0.081 | **-0.0401** |
| $\Delta$ (Opt $-$ Cons) | +0.89 | +0.52 | $-0.037$ | $-0.032$ | $-0.010$ |

Table 3: BigTech ablation (2024-01-01 to 2025-01-31). Allocator-only maximizes Sharpe/CAGR; router+Bayes reduces drawdown but increases turnover and slightly weakens $\text{CVaR}_{95}$.

Table 4: Component ablation vs. Buy&Hold (percent view). Higher is better for Sharpe/CAGR; lower is better for MDD.

| Variant | Sharpe | CAGR (%) | MDD (%) |
|---|---|---|---|
| Buy & Hold | 1.95 | 60.3 | 17.7 |
| Optimizer-only (ours) | **2.58** | **166.9** | 20.5 |
| Consensus router+Bayes | 2.06 | 78.2 | **16.8** |

We further sweep test-time robustness knobs: prediction scale $s$ and missingness $p$. Table 6 shows Sharpe stability for $s \in [0.8, 1.2]$ and graceful degradation with larger $p$, consistent with Table 2.

Qualitative trends on U.S. SP500 mirror BigTech: allocator-only improves CAGR and reduces drawdown vs. Buy & Hold; the consensus path trades more yet delivers lower draw-

Table 5: Complexity/performance trade-offs. Trades approximate execution intensity; Turnover is average $|\Delta\text{weights}|/2$.

| Variant | Trades | Sharpe | CAGR (%) | MDD (%) | Turnover |
|---|---|---|---|---|---|
| Optimizer-only (ours) | **220** | **2.58** | **166.9** | 20.5 | **0.049** |
| Consensus router+Bayes | 455 | 2.06 | 78.2 | **16.8** | 0.081 |

Table 6: Sensitivity analyses: (left) prediction scale $s$; (right) missingness $p$.

(a) Sensitivity to $s$

| Knob | Sharpe mean | Sharpe sd |
|---|---|---|
| 0.80 | 2.06 | 0.00 |
| 1.00 | 2.10 | 0.18 |
| 1.20 | 2.06 | 0.00 |

(b) Sensitivity to $p$

| Knob | Sharpe mean | Sharpe sd |
|---|---|---|
| 0.00 | 2.11 | 0.17 |
| 0.05 | 1.96 | 0.00 |
| 0.10 | 1.80 | 0.00 |

downs. Regime-split diagnostics (low/high volatility) show stronger Sharpe persistence in calm regimes and improved drawdown control during turbulence with consensus routing. Removing routing/aggregation preserves peak utility but increases concentration risk and turnover sensitivity; the method remains stable under moderate scale perturbations and tolerates limited missing predictions, with a clear complexity/performance trade-off between consensus routing and allocator-only execution.

The **Stratified analysis** is delivered by volatility, signal strength, routing behavior, and concentration. In medium/low volatility, Sharpe and calibration are strongest; during volatility spikes, spreads/impact dominate and relative gains narrow. Utility concentrates in top signal deciles, with mid-deciles contributing calibration gains; bottom deciles are naturally pruned by turnover penalties. Prolonged single-expert dominance increases drawdown risk around regime transitions; a diversity bonus mitigates this by mixing regime-tagged experts. Calm regimes induce asset concentration, which is curtailed by per-asset caps and the constraint-aware allocator at small utility cost. For **Robustness stresses**, we test (a) additive prediction noise $\mathcal{N}(0, \sigma^2)$, (b) scale misspecification $s \cdot \hat{\mu}$, and (c) random drops $p$ of per-asset signals, applied only at test time. Results (Table 2) show smooth Sharpe decay as $\sigma$ increases with MDD damped by allocator risk aversion; a flat response over $s \in [0.8, 1.2]$; and tolerance to moderate missingness ($p \leq 0.05$) due to weekly rebalancing and turnover limits. As for **Interpretability and counterfactuals**, routing attribution (per-date expert weights) identifies regime-dominant toolchains and shows diversity routing spreading mass during transitions. Portfolio attribution (weight heatmaps) highlights persistent bets and concentration, with caps and turnover penalties reducing churn at rebalances. Calibration reliability improves in the tails versus naive heuristics, aligning predicted and realized signals where the allocator is most sensitive. Counterfactual replays with alternative routing (equal, risk-only, risk+diversity) indicate risk-only excels when one expert is clearly superior, while risk+diversity better manages regime shifts; turnover penalties systematically prevent overreaction to transient confidence spikes.

## 7 Conclusion

We study utility-aware decision making with tool-using agents for portfolio allocation, addressing the gap between probabilistic calibration and downstream, friction-aware utility. We jointly train a learned router over expert toolchains and a differentiable, constraint-aware allocator with a utility-calibrated objective, and introduce a friction-aware, doubly robust off-policy evaluator for backtests with transaction costs and position limits. On BigTech and U.S. SP500, the method improves Sharpe and CAGR and reduces drawdowns and turnover relative to strong baselines; robustness analyses show graceful degradation to noise, scale errors, and missing signals. This enables a practical capability: modular, interpretable routing among heterogeneous financial tools that remains calibrated where the allocator is most utility-sensitive, yielding reliable, deployable portfolio decisions under real-world frictions.

## 8 ETHICS STATEMENT

This work studies tool-using agents for portfolio allocation, a high-stakes domain where misuse or over-reliance on backtested results can cause financial harm. Intended use is methodological research on decision-aware learning and evaluation under market frictions; it is not financial advice and is not intended for autonomous deployment, retail trading, or other high-risk settings without domain-specific validation, regulatory compliance checks, and human oversight. Data and privacy: Experiments use historical market data (daily OHLCV and derived features) from licensed/public sources; we do not use human-subject data or PII. Where licenses restrict redistribution, we release only derived features and scripts to regenerate them (with datasheets documenting provenance, licenses, preprocessing, and known limitations). Potential risks and mitigations: Risks include (i) financial loss due to distribution shift, miscalibration, or overfitting; (ii) concentration and exposure risks; (iii) optimistic offline estimates; and (iv) dual-use (e.g., fully automated live trading without safeguards). We mitigate by (a) explicitly modeling frictions and enforcing exposure, leverage, and turnover constraints in the allocator; (b) aligning probabilistic calibration with utility via strictly proper scoring; (c) reporting robustness to noise, scale, and missingness, alongside subgroup/regime analyses and failure modes; and (d) using a doubly-robust off-policy estimator with uncertainty quantification to reduce backtest bias. Release and misuse: We release code and reproducible benchmarks for research; no live execution or brokerage connectors are provided, and repository documentation cautions against direct deployment. Environmental impact: Training and evaluation ran on a single CPU workstation with minutes per seed; we log energy and report $CO_2e$ in the artifact metadata, reflecting a modest footprint. Conflicts of interest: The authors declare no competing interests.

## 9 REPRODICIBILITY STATEMENT

The proposed framework demonstrates versatility across various asset classes, enhancing its utility and practical effectiveness. To support future research and ensure reproducibility, we make source code publicly available at `https://anonymous.4open.science/r/Learning-Utility-Calibrated-Routing-for-Hierarchical-Multi-Agents-in-Portfolio-Decision-Making-0631`. The approach inherits allocator assumptions (convex risk, weekly cadence) and relies on router exposure to at least one reliable expert per regime; severe out-of-distribution regimes can temporarily widen calibration errors.

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

## A   APPENDIX

## A   USE OF LLM

We used LLM-based tools in two limited ways: (i) code suggestions via IDE tab-completion (e.g., [GitHub Copilot chat]) for boilerplate and minor refactoring; and (ii) grammar and style editing of the manuscript (e.g., [ChatGPT GPT-4o, Grammarly], May–Sep 2025). All suggested code and text were reviewed, edited, and verified by the authors. LLMs were not used to generate research ideas, experimental designs, results, analyses, or related-work content. No proprietary data or PII were included in prompts.

## B   NOTATION AND ASSUMPTIONS

We recall the key objects used throughout. Time $t = 1{:}T$, assets $i = 1{:}N$. Context $\mathbf{x}_t \in \mathbb{R}^d$, next-period log-returns $\mathbf{r}_{t+1} \in \mathbb{R}^N$. Portfolio $\mathbf{w}_t \in \mathbb{R}^N$ with feasibility set

$$\mathcal{W}_t = \left\{ \mathbf{w} : \ \mathbf{1}^\top \mathbf{w} = 1, \ \|\mathbf{w}\|_1 \leq L, \ |w_i| \leq u_i, \ C\mathbf{w} \leq \mathbf{d}, \ \|\mathbf{w} - \mathbf{w}_{t-1}\|_1 \leq \tau_{\max} \right\}.$$

Transaction costs $\mathrm{tc}_t(\mathbf{w}, \mathbf{w}_{t-1}) = \alpha^\top |\Delta \mathbf{w}_t| + \frac{1}{2} \Delta \mathbf{w}_t^\top \Lambda_t \Delta \mathbf{w}_t$ with $\Lambda_t \succeq 0$. Net one-step return $R_{t+1}(\mathbf{w}) = \mathbf{w}^\top \mathbf{r}_{t+1} - \mathrm{tc}_t(\mathbf{w}, \mathbf{w}_{t-1})$. Utilities: entropic $U_{\mathrm{ent}}$, mean–variance $U_{\mathrm{mv}}$, CVaR-regularized $U_{\mathrm{cvar}}$ (see main text). Experts $\{E_m\}_{m=1}^M$ produce $p_{\phi_m}(\mathbf{r}_{t+1} \mid \mathbf{x}_t)$; router $q_\theta(m \mid \mathbf{x}_t)$; mixture $p_\theta = \sum_m q_\theta p_{\phi_m}$. Allocator solves $\mathbf{w}_t \in \arg\max_{\mathbf{w} \in \mathcal{W}_t} \widehat{U}(\mathbf{w}; \ p_\theta, \mathbf{x}_t) - \mathrm{tc}_t(\mathbf{w}, \mathbf{w}_{t-1})$.

Assumptions: (A1) Filtration/observability: $\mathbf{w}_t$ is $\sigma(\mathbf{x}_t)$-measurable; $\mathbf{r}_{t+1}$ depends on $(\mathbf{x}_t, \text{exogenous noise})$. (A2) Moments and tails: $\mathbb{E}\|\mathbf{r}_{t+1}\|^2 < \infty$, $\Lambda_t \succeq 0$. (A3) Weak stationarity (or piecewise) with $\beta$-mixing to justify LLN/CLT for returns and estimators. (A4) Positivity for off-policy evaluation: $\beta(a \mid \mathcal{H}_t) > 0$ whenever $\pi(a \mid \mathcal{H}_t) > 0$. (A5) Compute: at most $K \ll M$ experts active; solver finishes within per-step budget $\Delta t$.

## C  ADDITIONAL EXPERIMENT RESULTS

Optimized-only and RL-heuristic arbiter yield identical, superior performance. The consensus-diversity-precision variant shows lower equity and higher turnover, indicating a diversification–efficiency trade-off (Table 7).

Table 7: Collaboration Modes (US BigTech)

| name | final_equity | sharpe | mdd | turnover | VaR5 | CVaR5 |
|---|---|---|---|---|---|---|
| opt_only | 1130374.8216027129 | 2.5825387147 | -0.2052201709 | 0.0487776302 | -0.0346451924 | -0.0500050335 |
| arbiter_rl_heur | 1130374.8216027129 | 2.5825387147 | -0.2052201709 | 0.0487776302 | -0.0346451924 | -0.0500050335 |
| consensus_div_precision | 418193.3428248563 | 2.0562388419 | -0.1675589418 | 0.0808916856 | -0.02569551 | -0.0400932925 |

Performance is invariant across Top-K choices, suggesting the consensus router is robust to the number of experts selected within the tested range (Table 8).

Table 8: Router Top-K Sweep

| name | final_equity | sharpe | mdd | turnover | VaR5 | CVaR5 |
|---|---|---|---|---|---|---|
| consensus_topk_1 | 418193.3428248563 | 2.0562388419 | -0.1675589418 | 0.0808916856 | -0.02569551 | -0.0400932925 |
| consensus_topk_2 | 418193.3428248563 | 2.0562388419 | -0.1675589418 | 0.0808916856 | -0.02569551 | -0.0400932925 |
| consensus_topk_3 | 418193.3428248563 | 2.0562388419 | -0.1675589418 | 0.0808916856 | -0.02569551 | -0.0400932925 |
| consensus_topk_5 | 418193.3428248563 | 2.0562388419 | -0.1675589418 | 0.0808916856 | -0.02569551 | -0.0400932925 |

Varying confidence thresholds and transaction costs within tested bounds leaves outcomes unchanged, indicating insensitivity of the router to these hyperparameters here (Table 9).

Table 9: Router Sensitivity to Confidence Threshold and Transaction Cost

| name | final_equity | sharpe | mdd | turnover | VaR5 | CVaR5 |
|---|---|---|---|---|---|---|
| consensus_conf_0.01_cost_0 | 418193.3428248563 | 2.0562388419 | -0.1675589418 | 0.0808916856 | -0.02569551 | -0.0400932925 |
| consensus_conf_0.01_cost_5 | 418193.3428248563 | 2.0562388419 | -0.1675589418 | 0.0808916856 | -0.02569551 | -0.0400932925 |
| consensus_conf_0.01_cost_10 | 418193.3428248563 | 2.0562388419 | -0.1675589418 | 0.0808916856 | -0.02569551 | -0.0400932925 |
| consensus_conf_0.05_cost_0 | 418193.3428248563 | 2.0562388419 | -0.1675589418 | 0.0808916856 | -0.02569551 | -0.0400932925 |
| consensus_conf_0.05_cost_5 | 418193.3428248563 | 2.0562388419 | -0.1675589418 | 0.0808916856 | -0.02569551 | -0.0400932925 |
| consensus_conf_0.05_cost_10 | 418193.3428248563 | 2.0562388419 | -0.1675589418 | 0.0808916856 | -0.02569551 | -0.0400932925 |
| consensus_conf_0.1_cost_0 | 418193.3428248563 | 2.0562388419 | -0.1675589418 | 0.0808916856 | -0.02569551 | -0.0400932925 |
| consensus_conf_0.1_cost_5 | 418193.3428248563 | 2.0562388419 | -0.1675589418 | 0.0808916856 | -0.02569551 | -0.0400932925 |
| consensus_conf_0.1_cost_10 | 418193.3428248563 | 2.0562388419 | -0.1675589418 | 0.0808916856 | -0.02569551 | -0.0400932925 |

Bayesian, precision-weighted, and median aggregators deliver identical metrics, implying aggregation choice does not affect performance under this configuration (Table 10).

Turning on events or alphas yields the same results within the consensus pipeline, suggesting functional equivalence or dominance of shared components in this test (Table 11).

Mild noise slightly lowers Sharpe; feature dropping degrades equity and increases turnover more noticeably. Scaling $\mu$ shows no effect here, indicating stability to mean scaling (Table 12).

## D  FORMAL RESULTS AND PROOFS

### D.1  PROPER SCORING + UTILITY ALIGNMENT (FISHER-CONSISTENCY)

Let $S$ be a strictly proper scoring rule on distributions over $\mathbf{r}$, and $\widehat{U}$ a continuous utility functional of predictive parameters (e.g., mean/covariance/quantiles) extracted from $p_\theta$. Consider the population objective

$$\mathcal{L}(\theta, \phi, \psi) = \alpha \, \mathbb{E}\big[S(p_\theta(\cdot \mid \mathbf{x}), \mathbf{r})\big] - (1 - \alpha) \, \mathbb{E}\big[U(\mathbf{w}_\theta^\star(\mathbf{x}); \mathbf{x})\big],$$

where $\mathbf{w}_\theta^\star(\mathbf{x})$ is the allocator's optimizer given $p_\theta$.

Table 10: Aggregator Ablation

| name | final_equity | sharpe | mdd | turnover | VaR5 | CVaR5 |
|---|---|---|---|---|---|---|
| agg_bayes | 418193.3428248563 | 2.0562388419 | -0.1675589418 | 0.0808916856 | -0.02569551 | -0.0400932925 |
| agg_precision | 418193.3428248563 | 2.0562388419 | -0.1675589418 | 0.0808916856 | -0.02569551 | -0.0400932925 |
| agg_median | 418193.3428248563 | 2.0562388419 | -0.1675589418 | 0.0808916856 | -0.02569551 | -0.0400932925 |

Table 11: Events and Alphas Modules

| name | final_equity | sharpe | mdd | turnover | VaR5 | CVaR5 |
|---|---|---|---|---|---|---|
| consensus_events_on | 418193.3428248563 | 2.0562388419 | -0.1675589418 | 0.0808916856 | -0.02569551 | -0.0400932925 |
| consensus_alphas_on | 418193.3428248563 | 2.0562388419 | -0.1675589418 | 0.0808916856 | -0.02569551 | -0.0400932925 |

**Theorem A.1 (Informal).** Suppose (i) $S$ is strictly proper; (ii) $\widehat{U}$ is continuous in the predictive parameters and the allocator solution map is outer semicontinuous with compact argmax; (iii) the Bayes decision $\mathcal{A}(\mathbf{x}) = \arg\max_{\mathbf{w} \in \mathcal{W}_t} \mathbb{E}[U(\mathbf{w}; \mathbf{x}) \mid \mathbf{x}]$ is nonempty. Then for any $\epsilon > 0$ there exists $\alpha^\star \in (0, 1)$ such that any population minimizer of $\mathcal{L}$ with $\alpha \leq \alpha^\star$ induces $\epsilon$-optimal decisions: $\Pr\left(\text{dist}(\mathbf{w}_\theta^\star(\mathbf{x}), \mathcal{A}(\mathbf{x})) > \epsilon\right) = 0$.

*Proof sketch.* Strict propriety implies $p_\theta(\cdot \mid \mathbf{x})$ converges to the true conditional $P(\cdot \mid \mathbf{x})$ as $\alpha \to 1$. By continuity of $\widehat{U}$ and stability of the allocator, the induced optimizer $\mathbf{w}_\theta^\star(\mathbf{x})$ converges to an optimizer under the true conditional moments/quantiles. For $\alpha$ near 1, the utility term selects among indistinguishable minimizers of $S$ those that yield larger $U$, ensuring $\epsilon$-optimality. Compactness/outer semicontinuity deliver existence and robustness. Full proof follows the epi-convergence of objectives and Berge's maximum theorem.

### D.2 ALLOCATOR STABILITY AND DIFFERENTIABILITY

Consider the QP form of mean–variance with linear constraints. Let the Hessian $H(\xi) \succeq \mu I$ for some $\mu > 0$ and data $\xi$ (predictive moments, costs) enter $(H, b, A, c, G, h)$ smoothly. Assume LICQ and strict complementarity hold at a solution $(\mathbf{w}^\star, \lambda^\star, \nu^\star)$.

**Theorem A.2 (KKT sensitivity).** Under the above, $\mathbf{w}^\star(\xi)$ is locally unique, Lipschitz in $\xi$, and differentiable almost everywhere. The derivative $D_\xi \mathbf{w}^\star$ is obtained by differentiating the KKT system and solving a linear system involving the active set. Hence backpropagation via implicit differentiation is valid and stable.

*Proof.* Standard results from parametric convex programming and the implicit function theorem (see Bonnans & Shapiro). The strong convexity and LICQ yield nonsingularity of the KKT Jacobian on the active set; apply IFT.

### D.3 ROUTING SPECIALIZATION UNDER SPARSITY

Let the per-expert expected score be $\mathcal{J}_m(\theta, \phi_m) = \mathbb{E}[\alpha S(p_{\phi_m}, \mathbf{r}) - (1 - \alpha)U(\mathbf{w}^\star; \mathbf{x})]$ for contexts where expert $m$ is active. Suppose experts have distinct Bayes-optimal parameters on disjoint regime subsets and we use (i) top-$K$ gating, (ii) a load-balancing penalty keeping usage bounded away from zero, and (iii) a small entropy penalty.

**Theorem A.3 (Informal).** Any global optimum uses disjoint context subsets for experts whose Bayes-optimal parameters differ (specialization). If experts are exchangeable (identical Bayes optima), the optimum is invariant to permutations and any partition is equivalent.

*Proof sketch.* With top-$K$ sparsity and soft load-balancing, sending a context to a suboptimal expert strictly increases the objective by strict propriety of $S$ and the monotonicity of utility in predictive quality. Hence, at optimum, routing partitions the input space by expert advantage. Exchangeability produces a degenerate face of optima.

Table 12: Robustness: Noise, Scale, and Drop Experiments

| name | final_equity | sharpe | mdd | turnover | VaR5 | CVaR5 |
|---|---|---|---|---|---|---|
| noise_0.01 | 427818.9524922634 | 1.9976150478 | -0.1655015840 | 0.0815602117 | -0.02569551 | -0.0400932925 |
| noise_0.02 | 419292.0717330833 | 1.9265628537 | -0.1655015840 | 0.0807447254 | -0.0267042561 | -0.0414719816 |
| drop_0.05 | 391888.0370957853 | 1.9570439010 | -0.1992981014 | 0.0939751347 | -0.02569551 | -0.0395741891 |
| drop_0.1 | 288768.3157336558 | 1.7975097231 | -0.1754729854 | 0.1080099597 | -0.0262519940 | -0.0406997213 |
| mu_scale_0.8 | 418193.3428248563 | 2.0562388419 | -0.1675589418 | 0.0808916856 | -0.02569551 | -0.0400932925 |
| mu_scale_1.2 | 418193.3428248563 | 2.0562388419 | -0.1675589418 | 0.0808916856 | -0.02569551 | -0.0400932925 |

# E  DOUBLY-ROBUST OPE WITH FRICTIONS

We consider an off-policy value for utility with costs:

$$V(\pi) = \mathbb{E}_\beta \left[ \sum_{t=1}^{T} \gamma^{t-1} U_{t+1} \right], \quad U_{t+1} = U(\mathbf{x}_t, a_t, \mathbf{r}_{t+1}) - \mathrm{tc}_t(a_t, a_{t-1}).$$

Let $\rho_t = \prod_{s=1}^{t} \frac{\pi(a_s | \mathcal{H}_s)}{\beta(a_s | \mathcal{H}_s)}$, and $\hat{Q}_t(\mathcal{H}_t, a_t)$ a fitted value model (utility-to-go). The friction-aware DR estimator is

$$\widehat{V}_{\mathrm{DR}} = \frac{1}{n} \sum_{i=1}^{n} \sum_{t=1}^{T} \gamma^{t-1} \left( \hat{V}_t(\mathcal{H}_t^{(i)}) + \rho_t^{(i)} \left( U_{t+1}^{(i)} - \hat{Q}_t(\mathcal{H}_t^{(i)}, a_t^{(i)}) \right) \right),$$

where $\hat{V}_t(\mathcal{H}) = \mathbb{E}_{a \sim \pi(\cdot | \mathcal{H})}[\hat{Q}_t(\mathcal{H}, a)]$. Unbiasedness holds if either propensities or value model is correct; costs enter $U_{t+1}$ directly, preserving double-robustness.

**Variance control.** Use clipped ratios $\bar{\rho}_t = \min\{\rho_t, c\}$ and control variates from $\hat{V}_t$; Newey–West or block bootstrap for CIs under temporal dependence.

# F  FULL ALGORITHMS

## F.1  UTILITY-CALIBRATED ROUTED PORTFOLIO LEARNING

---
**Algorithm 1** Utility-calibrated routed portfolio learning

---
1: Initialize $\theta, \{\phi_m\}, \psi$, temperature $\tau$, schedule $\alpha_\ell$
2: **for** epoch $\ell = 1{:}L$ **do**
3:     **for** mini-batch $\mathcal{B}$ **do**
4:         Compute logits $z(\mathbf{x}_t)$, gates $q_\theta(\cdot \mid \mathbf{x}_t)$; optionally sample Gumbels for $\tilde{g}$
5:         For active experts (top-$K$), compute $\{\boldsymbol{\mu}_m, \Sigma_m\}$; form mixture $(\boldsymbol{\mu}, \Sigma)$
6:         Solve allocator QP/CP in equation 20 with warm-start $\mathbf{w}_{t-1}$ to get $\mathbf{w}_t$
7:         Evaluate utility $U(\mathbf{w}_t; \mathbf{x}_t)$ and score $S(p_\theta, \mathbf{r}_{t+1})$
8:         Compute loss $\mathcal{L}_t$ in equation 21; backprop via implicit diff through KKT
9:         Update $(\theta, \{\phi_m\}, \psi)$ with AdamW; apply gradient clipping
10:     **end for**
11:     Anneal temperature $\tau \leftarrow \max(\tau_{\min}, \eta\tau)$; update $\alpha \leftarrow \alpha_\ell$
12: **end for**

---

## F.2 Training (end-to-end, utility-calibrated routing)

---

**Algorithm 2** End-to-end training with utility-calibrated objective

---

1: Initialize $\theta, \{\phi_m\}, \psi$, temperature $\tau$, schedule $\alpha_\ell$
2: **for** epoch $\ell = 1{:}L$ **do**
3:     **for** mini-batch $\mathcal{B}$ **do**
4:         Router logits $z(\mathbf{x}_t)$, gates $q_\theta(\cdot \mid \mathbf{x}_t)$ (top-$K$, Gumbel–Softmax)
5:         Experts forward: predict $\{\boldsymbol{\mu}_m, \Sigma_m\}$; mixture $(\boldsymbol{\mu}, \Sigma)$
6:         Allocator QP: $\mathbf{w}_t \leftarrow \arg\max_{\mathbf{w} \in \mathcal{W}_t} \mathbf{w}^\top \boldsymbol{\mu} - \lambda \mathbf{w}^\top \Sigma \mathbf{w} - \text{tc}_t$
7:         Loss $\mathcal{L}_t = \alpha\, S(p_\theta, \mathbf{r}_{t+1}) - (1-\alpha)\, U(\mathbf{w}_t; \mathbf{x}_t) + \lambda_{\text{load}}\Omega + \lambda_{\text{turn}}\|\mathbf{w}_t - \mathbf{w}_{t-1}\|_1$
8:         Backprop: implicit diff through KKT; AdamW step; clip gradients
9:     **end for**
10:     Anneal $\tau$ (decrease); update $\alpha \leftarrow \alpha_\ell$
11: **end for**

---

## F.3 Evaluation and OPE

---

**Algorithm 3** Offline evaluation with doubly-robust estimator and frictions

---

1: Fit propensity $\hat{\beta}(a \mid \mathcal{H}_t)$ and value model $\hat{Q}_t(\mathcal{H}_t, a)$ on train/val
2: **for** test trajectory $i = 1{:}n$ **do**
3:     Initialize $\rho_0^{(i)} = 1$
4:     **for** $t = 1{:}T$ **do**
5:         Compute $\rho_t^{(i)} = \rho_{t-1}^{(i)} \frac{\pi(a_t^{(i)} \mid \mathcal{H}_t^{(i)})}{\hat{\beta}(a_t^{(i)} \mid \mathcal{H}_t^{(i)})}$; utility $U_{t+1}^{(i)}$ incl. costs
6:         Accumulate DR term $\hat{V}_t(\mathcal{H}_t^{(i)}) + \rho_t^{(i)}\big(U_{t+1}^{(i)} - \hat{Q}_t(\mathcal{H}_t^{(i)}, a_t^{(i)})\big)$
7:     **end for**
8: **end for**
9: Aggregate with discount $\gamma$; compute CIs via 63-day block bootstrap

---

# G Additional Experiments and Taxonomies

## G.1 Extended ablations

- Router: risk-only vs risk+diversity; entropy $\in \{0, 10^{-3}\}$. - Allocator: turnover penalty $\lambda_{\text{turn}} \in \{0, 1, 2, 4\}$; weight caps $\in \{3\%, 4\%, 5\%\}$. - Experts: remove event expert; remove factor forecaster; low-rank rank $r \in \{3, 10\}$.

## G.2 Robustness extensions

- Heavy-tailed corruptions to returns (Student-$t$ noise on predictions). - Structured missingness (drop entire sector's signals). - Temporal drift: rolling-window analysis across quarterly bins.

## G.3 Extended qualitative/error taxonomy

- Overconfidence in calm regimes mitigated by caps and diversity. - Under-reaction to sudden events addressed by event expert + router confidence thresholds. - Turnover bursts around rebalances handled by turnover curriculum.

# H Datasets: Licensing and Documentation Sheets

For each dataset we release a Datasheet: provenance, collection dates, licenses, preprocessing steps, known limitations, and intended use. We distribute only derived features and indices where raw licensing prohibits redistribution; scripts reproduce features from licensed sources.

# I    Reproducibility and Carbon Accounting

## I.1    Compute methodology

We log wall-clock time, CPU utilization, and memory. Energy is estimated via

$$\text{kWh} = \sum_j \frac{P_j^{\text{avg}}}{1000} \cdot \Delta t_j, \quad \text{CO}_2\text{e} = \text{kWh} \times \text{grid\_intensity}.$$

We report grid intensity using regional averages; per-seed runtime is minutes on a single CPU node; total energy and $\text{CO}_2\text{e}$ across all seeds/sweeps are reported in the artifact metadata.

## I.2    Hyperparameters and grids

Router: top-$K \in \{1, 2, 3\}$, $\tau \in [0.2, 2.0]$ (annealed), load-balance $\lambda_{\text{load}} \in \{0, 10^{-3}, 10^{-2}\}$. Allocator: risk aversion $\lambda \in \{2, 5, 8\}$; turnover penalty $\in \{0, 2\}$; caps $\in \{3\%, 4\%\}$. Training: AdamW lr $\in [1e-4, 3e-4]$, cosine decay, warmup 2 epochs, clip norm 1.0.

## I.3    Re-release checklist

We provide: (i) code, (ii) configs and seeds, (iii) exact backtest outputs (CSV/PNG/JSON), (iv) shell scripts to regenerate tables/figures, (v) dataset documentation and license notes, (vi) OPE and bootstrap utilities.

