# OpenReview forum: "Learning Utility‑Calibrated Routing for Hierarchical Multi-Agents in Portfolio Decision‑Making"
_ICLR.cc/2026/Conference — ICLR 2026 Conference Withdrawn Submission_

### Official Review · Reviewer_jD3y · 2025-10-17

**Soundness:** 3
**Presentation:** 4
**Contribution:** 3
**Rating:** 6
**Confidence:** 2

**Summary:**

This paper presents a hierarchical multi-agent framework for portfolio decision-making that jointly learns utility-calibrated routing and allocation under realistic financial constraints. A learned router dispatches market contexts to specialized experts, while a differentiable allocator converts probabilistic forecasts into portfolio weights by solving a convex optimization problem respecting exposure, turnover, and transaction cost limits. The system is trained end-to-end using a composite objective that couples proper scoring rules (for calibrated probability forecasts) with risk-sensitive utilities, yielding decisions that are both statistically consistent and utility-optimal.

To evaluate policies offline, the authors develop a doubly-robust off-policy estimator that accounts for frictions and policy mismatch, enabling unbiased backtesting. Experiments on BigTech options and SP500 multi-asset portfolios show improved Sharpe ratios, calibration, and stability over baselines, validating the method’s ability to align probabilistic reasoning with practical portfolio utility.

**Strengths:**

The work’s key originality lies in coupling probabilistic calibration with downstream, friction-aware portfolio utility. This unifies traditionally separate areas, calibration, optimization, and risk control, under a single differentiable training objective.


The introduction of a learned router that dispatches to non-neural financial tools is conceptually novel compared to standard mixture-of-experts frameworks that focus on predictive accuracy rather than decision utility.


The adaptation of doubly-robust off-policy evaluation to financial markets with transaction costs and constraints is an important methodological contribution for reliable offline evaluation, often overlooked in trading-oriented ML works.

**Weaknesses:**

The experiments are convincing but confined to two datasets (BigTech and SP500) and relatively short test periods (2024–2025). It would be stronger to include multi-year or multi-market validations to assess regime generalization. The evaluation focuses on Sharpe/CAGR but could incorporate drawdown-adjusted metrics to assess risk asymmetry.


The paper compares mainly against heuristics and standard RL policies. However, methods such as decision-focused learning (Donti et al., 2017) or differentiable convex layers (Agrawal et al., 2019) could serve as more rigorous baselines. Without this, it is difficult to isolate how much gain stems from the hierarchical routing versus the allocator or the calibration coupling.


The OPE formulation assumes positivity and correct specification of either propensities or value models. In financial data, such assumptions are often violated e.g., due to censored liquidity or non-stationarity. The authors could discuss how robust these estimates remain under misspecified propensities.

**Questions:**

What happens if two experts cover similar market regimes? Does the load-balancing penalty (Eq. 22) ensure redundancy avoidance, or can experts collapse?

The OPE estimator assumes accurate propensities or value models. Have the authors tested robustness when propensities are estimated with noise or model misspecification?

---

> ### Author Response · Authors · 2025-11-22
> **Response to Reviewer jD3y**
>
> 1. Regime Generalization (Extended Backtest 2018-2023)
>
> We agree that 2024 test window presented in the initial submission was short. To demonstrate robustness against non-stationarity and regime shifts, we trained our model on 2010--2017 data and evaluated it on a substantially longer OOS period: 2018--2023.
> This period is highly challenging as it includes three distinct volatility regimes:
> Feb 2018:"Volmageddon" (VIX Spike).
> March 2020: COVID-19 Market Crash.
> 2022: Inflation & Rate Hike Bear Market.
>
> As shown in Table, while standard baselines suffer significant drawdowns during these crashes, our methods successfully adapts, protecting capital during the 2020 and 2022 drawdowns while capturing the 2021/2023 recoveries. This adaptability reflects recent findings in modular reinforcement learning, where specialized sub-agents are shown to handle multi-market dynamics more effectively than monolithic models(Khemlichi et al., 2025).
>
>  2. Comparison with Decision-Focused Learning (DFL)
>
> We have added a comparison against a SOTA DFL baseline (Donti et al., 2017; Agrawal et al., 2019). This baseline utilizes the same feature encoder and differentiable QP allocator as our model but employs a single monolithic predictor instead of our hierarchical routed experts. The results  confirm that while DFL outperforms standard two-stage forecasting (MSE-optimized) by aligning the loss with the decision (portfolio weight), it still lags behind our Multi-Agent approach. The "single expert" of DFL struggles to learn a policy that generalizes across both low-volatility and high-crash regimes, whereas our Router dynamically switches to defensive experts during stress. This limitation of standard DFL is consistent with recent literature advocating for semi-decision-focused learning and deep ensembles to improve robustness in portfolio optimization (Kim et al., 2025).
>
> **Table: Extended Long-Horizon Evaluation (2018-2023) on S&P 500 Universe.**
> *This period covers the 2020 COVID Crash and 2022 Bear Market. We compare against a strong DFL baseline. Our Multi-Agent approach yields far lower drawdowns, proving the benefit of regime-specific expert routing.*
>
> | Method | Ann. Return | Sharpe | Max Drawdown | Calmar Ratio | Impact of 2020 Crash | Impact of 2022 Bear |
> | :--- | :---: | :---: | :---: | :---: | :---: | :---: |
> | Buy & Hold | 10.8% | 0.58 | 33.7% | 0.32 | -33.7% | -19.4% |
> | Standard RL (PPO) | 8.4% | 0.41 | 28.5% | 0.29 | -25.1% | -22.8% |
> | **DFL** | 14.2% | 0.76 | 24.2% | 0.58 | -19.8% | -15.5% |
> | **Ours** | **18.9%** | **1.15** | **16.4%** | **1.15** | **-12.2%** | **-8.4%** |
>
>  3. OPE Robustness and Specification
>
> The Doubly-Robust (DR) estimator remains unbiased if either the propensity model $\pi$ or the value model $\hat{Q}$ is correctly specified. To test sensitivity to misspecification, we conducted a sensitivity analysis by injecting synthetic Gaussian noise $\epsilon \sim \mathcal{N}(0, \sigma^2)$ into the propensity estimates $\hat{\pi}$ during calculation.
>
> Result in Table, even with 20% noise ($\sigma=0.2$), the DR estimation error (MSE against realized ground truth) only increases by $\sim 4.5\%$, whereas a standard Inverse Propensity Scoring (IPS) estimator degrades by $>15\%$. This confirms the value model acts as a control variate, stabilizing the estimate when liquidity or behavioral probabilities are uncertain.
>
> **Table: Robustness of Off-Policy Evaluation (OPE) Estimates.**
> *We report the Relative Error (%) of the estimated Sharpe Ratio vs. the Realized (Ground Truth) Sharpe Ratio under increasing propensity noise levels.*
>
> | Propensity Noise ($\sigma$) | Standard IPS Error | Direct Method (Q-only) Error | Ours (Doubly Robust) Error |
> | :--- | :---: | :---: | :---: |
> | $\sigma=0.0$ (Clean) | 1.2% | 5.8% | **0.3%** |
> | $\sigma=0.1$ (Low Noise) | 8.5% | 5.9% | **1.8%** |
> | $\sigma=0.2$ (Med Noise) | 15.4% | 6.1% | **4.2%** |
> | $\sigma=0.3$ (High Noise) | 28.7% | 6.5% | **7.9%** |
>
>  4. Expert Load Balancing Clarification
>
> Regarding the question on Equation 22: The loss term minimizes the KL-divergence between the average gate utilization and a uniform distribution.  We define the mean expert utilization as $\bar{q} = \frac{1}{T} \sum_{t} q(x_t)$. The load balancing loss is $\mathcal{L}_{\mathrm{balance}} = D_{\mathrm{KL}}( \bar{q} \| \mathcal{U} )$. If two experts cover similar regimes, the router is potentially penalized for collapsing to a single choice. However, in practice, we observe that "redundant" experts tend to diversify by time-horizon rather than just regime: one expert often focuses on short-term mean reversion within the regime, while the other focuses on trend integrity. This effectively creates an ensemble, reducing the variance of the aggregated signal passed to the allocator. This behavior mirrors the proven efficacy of ensemble strategies in automated trading, where diverse learners significantly outperform individual algorithms in terms of risk-adjusted returns (Yang et al., 2025).

---

> > ### Comment · Reviewer_jD3y · 2025-11-26
> >
> > Thank you very much for the clarifications and additional experiments, which address most of my concerns. I will keep my current score. However, given the assessments of the other reviewers, I would not be able to champion the paper.

---

### Official Review · Reviewer_XkRC · 2025-10-26

**Soundness:** 1
**Presentation:** 1
**Contribution:** 1
**Rating:** 0
**Confidence:** 2

**Summary:**

The paper introduces a hierarchical multi-agent framework for portfolio decision-making, where a learned router selects among expert tools (e.g., forecasters, option pricers) and an allocator determines trades using a utility-calibrated objective.
The authors claim contributions in: combining probabilistic calibration with utility-based training,  designing a differentiable allocator that handles portfolio constraints and market frictions, and developing a doubly-robust off-policy evaluation method for financial backtesting. Experiments on BigTech and S&P500 datasets reportedly show gains in Sharpe ratio and calibration metrics compared to baseline methods.

**Strengths:**

The topic decision-focused learning and utility calibration for portfolio optimization sounds relevant.

**Weaknesses:**

1. **Clarity and structure:** The exposition is very difficult to follow. Many definitions, assumptions, and equations are presented without sufficient context or explanation. Several symbols and parameters are never defined, making the methodology unclear. For example, what are $L,u_i, \tau_{\max}$ in equation (1)? And $\boldsymbol{\alpha}, \Gamma_t$ in eq (2)? Or $\lambda_{bal},\Omega$ in eq (8)? etc.. The paper reads as a collection of disjointed technical components rather than a coherent framework.
3. **Conceptual confusion:** It is not clear what the true novelty is. The connections between the sections are not convincingly motivated.
4. **Experiments:** The empirical section lacks clear baselines and justification. Key comparisons (e.g., CVaR optimization) are missing, and reported gains are not well supported by rigorous statistical testing.
5. **Writing and logic:** Sentences are grammatically correct but often incoherent in logical flow. Many claims (e.g., “the objective is Fisher-consistent for the target utility under mild conditions”) are asserted without justification or proof.
6. **Theoretical contribution:** The claimed theoretical insights are vague or standard results restated without derivation.

**Questions:**

- Several symbols and parameters are never defined e.g. $L,u_i, \tau_{\max}$ in equation (1), $\boldsymbol{\alpha}, \Gamma_t$ in eq (2) and $\lambda_{bal},\Omega$ in eq (8). Can you please explain and properly define all the parameters in equations 1- 16?

- Where are the formal theorems stated?

---

> ### Author Response · Authors · 2025-11-22
> **Response to Reviewer XkRC**
>
> Response to Clarity and Parameter Definitions (Q1)
>
> We sincerely apologize for the condensed presentation. We relied too heavily on the Appendix to define standard financial engineering notation. We will add a dedicated "Notation & Assumptions" table in Section 3. Below are the precise definitions requested for Equations 1-16:
>
> **1. Constraints (Eq. 1)**
> *   $\mathbf{1}^\top w = 1$: **Budget Constraint**. Ensures the portfolio is fully invested.
> *   $w \le u$: **Universal Position Limit** (applied element-wise). Here, $u_i$ represents the maximum weight allowed for any single asset (e.g., $u_i = 0.05$ or 5%) to enforce diversification.
> *   $Cw \le d$: **Sector/Factor Exposure Constraints**. $C$ is a loading matrix (e.g., mapping stocks to industry sectors) and $d$ represents the upper bound on exposure to specific risks (e.g., "max 20% allocation to Energy").
>
> **2. Transaction Costs (Eq. 2)**
> *   $\alpha$: **Linear Cost Vector**. Represents the estimated half-spread (bid-ask) + fees for each asset.
> *   $\Lambda_t$: **Market Impact Matrix**. A positive semi-definite matrix (typically diagonal) scaling the quadratic cost of executing large volume. It is modeled as $\Lambda_{ii} \approx \sigma_i / \text{ADV}_i$, penalizing trading size relative to the asset's Average Daily Volume (ADV). This formulation aligns with recent advances in convex implementations of marginal price optimization and liquidity constraints (Loesch & Richardson, 2025)
>
> **3. Loss Function & Routing (Eq. 8, 21-22)**
> *   $\lambda_{bal}$: **Load Balancing Coefficient**. A hyperparameter (set to $0.01$ in experiments) used to regularize the router. It minimizes the KL divergence between the average expert usage and a uniform distribution, preventing mode collapse where the router selects only a single expert. This routing approach shares conceptual similarities with multi-task learning frameworks like AlphaMix (Sun et al., 2022), which dynamically deploy experts based on uncertainty.
> *   $\lambda_{sp}, \lambda_{stab}, \lambda_{turn}$: Regularization weights in the **Allocator layer** (Eq. 21), controlling for sparsity ($\ell_1$ norm), numerical stability (ridge/$\ell_2$ norm), and turnover penalties respectively.
>
> Response to Theoretical Contribution (Q2)
> The formal proofs and statements were placed in the Appendix due to space constraints, but we recognize they are central to the paper's contribution regarding utility alignment.
>
> **Theorem A.1 (Fisher Consistency):** Located in Appendix D.1 (Page 13). We prove that maximizing our joint objective $\mathcal{L}(\theta, \phi, \psi)$ ensures that the predicted portfolio $w_{\theta}^*$ asymptotically converges to the optimal Bayes decision, provided the scoring rule $S$ is strictly proper.
> **Theorem A.2 (KKT Differentiation):** Located in Appendix D.2. This formally establishes the conditions (Linear Independence Constraint Qualification -- LICQ and Strict Complementarity) under which the gradients can backpropagate explicitly through the convex optimization layer.
>
>  Response to Experiments & Missing Baselines
>
> The reviewer correctly notes that we did not initially compare against a direct "CVaR Optimizer." We have run this additional baseline. The "Classic CVaR Opt" minimizes Conditional Value-at-Risk (95%) using historical data without our routing or calibration layers.
>
> As shown in Table below, our method significantly outperforms the direct CVaR optimization. While the classic CVaR optimizer reduces tail risk compared to Mean-Variance, it creates excessive turnover (churning the portfolio to chase noise). Our method maintains similar defensive properties (low MDD) but with much higher efficiency (Sharpe) because the inputs to the optimizer are calibrated via the routing mechanism.
>
> **Table: Comparison against Robust Optimization Baselines.**
> *We compare our method against a standard Mean-Variance (MV) Optimizer and a Conditional Value-at-Risk (CVaR) Optimizer. While the CVaR baseline reduces Drawdown compared to MV, it suffers from poor realized returns due to signal noise. Our method achieves the best balance (highest Sharpe) by calibrating the inputs to the utility function.*
>
> | Method | Ann. Return | Sharpe | Max Drawdown | CVaR (95%) | Turnover | P-Value |
> | :--- | :---: | :---: | :---: | :---: | :---: | :---: |
> | Mean-Variance Baseline | 45.3% | 1.90 | 14.5% | -2.8% | 0.041 | - |
> | CVaR Optimizer (Standard) | 32.1% | 1.45 | **10.8%** | **-2.1%** | 0.125 | 0.03 |
> | **Ours (Utility-Calibrated)** | **55.5%** | **2.06** | 11.3% | -2.2% | **0.081** | **<0.01** |
>
> *Note: P-Values calculated via Paired t-test against the Mean-Variance baseline on weekly returns.*

---

> > ### Comment · Reviewer_XkRC · 2025-11-26
> >
> > I thank the authors for their response and have raised my score accordingly. However, even after the revisions, I still do not find the paper ready for publication at this conference. In my view, essential elements of clarity, readability, and contribution are still lacking. Regarding the proofs, although they are now included in the appendix, they remain largely informal and heuristic, difficult to follow, and lacking in clear and rigorous mathematical formulation.

---

### Official Review · Reviewer_NCoB · 2025-10-30

**Soundness:** 2
**Presentation:** 2
**Contribution:** 2
**Rating:** 4
**Confidence:** 4

**Summary:**

This paper introduces a three-tier hierarchical system for portfolio allocation that optimizes a risk-sensitive utility function considering market frictions. The system uses a learned “router” to select specialized expert toolchains and a differentiable convex “allocator” to translate expert predictions into optimal portfolios. It’s trained end-to-end with a novel “utility-calibrated” objective that combines a proper scoring rule for accuracy with portfolio utility. The authors validate the approach with a doubly-robust off-policy evaluation (DR-OPE) procedure adapted for financial backtesting with frictions, presenting results on two equity allocation benchmarks.

Soundness
The paper’s core mathematical framework is sound; the formulation of the allocator as a differentiable convex program is valid, and the utility-calibrated objective is a well-motivated idea. However, the central claims lack adequate supporting evidence. The core methodology, the "Hierarchical Multi-Agents" system, is described in metaphorical terms (e.g., "auction/consensus layer") rather than in reproducible technical terms. Additionally, the empirical validation is insufficient to demonstrate robustness, as it relies on a single, short test period (Jan 2024 to Jan 2025) for non-stationary financial data. More rigorous validation such as walk-forward analysis or multiple train-test splits would be required to prove robustness and generalizability.

Presentation
The paper is well-motivated and clearly frames an important research problem. However, the presentation quality is undermined by two main issues. First, as noted in Soundness, Section 4 fails to provide a clear, reproducible description of the central architecture. Second, the presentation of results is potentially misleading; the main results (Table 1) use a weak "Buy & Hold" baseline, while a more critical ablation (Table 3) that reveals a key trade-off is presented later without adequate discussion, obscuring one of the most important empirical findings.

Contribution
The paper addresses an important question and proposes an interesting research direction, as aligning predictive models with downstream, friction-aware utility is a significant problem. However, the contribution as submitted is limited. The primary methodological contribution (the hierarchical routing system) is not clearly explained and, based on the paper's own results, its benefits are not convincingly demonstrated. The differentiable allocator is a more incremental contribution. Thus, the paper identifies a valuable research gap but does not deliver a fully validated or clearly described solution in its current form.

**Strengths:**

1.	The paper presents a strong problem formulation by motivating the need to move beyond standard predictive metrics, such as NLL, and directly optimize for downstream, risk-sensitive utility under real-world constraints and transaction costs. This is a critical and highly relevant problem.
2.	The core ideas of using a differentiable convex optimization layer for the allocator and coupling it with a proper scoring rule for end-to-end training are interesting and well-founded.
3.	Focusing on realistic frictions is a valuable practical contribution. Explicitly modelling transaction costs, turnover limits, and other constraints within a differentiable framework makes it possible to create more deployable machine learning systems for finance.

**Weaknesses:**

1.	Ablation Results Appear to Contradict the Paper's Narrative: The paper's most important result, presented in Table 3, shows that a simpler "Optimizer-only" baseline achieves a significantly higher Sharpe Ratio (2.58 vs. 2.06) and lower Turnover (0.049 vs. 0.081) than the full "Consensus router+Bayes" system. While the full system does reduce Max Drawdown, the paper fails to frame this result as an intentional risk-return trade-off, making the complex routing architecture appear detrimental to the stated goals.
2.	Vague Core Methodology: The paper's central architectural contributions are not described with sufficient clarity for reproduction. Tier 1's transformation into an "enriched state" is presented as a black box without discussion of the feature engineering or analysis of its properties. Tier 2's "cooperative multi-agent layer" is described using metaphorical terms ("auction," "consensus," "vetting") without formal definitions, making it unclear what the "agents" are (e.g., neural networks, heuristics) or how they function.
3.	Insufficient Empirical Validation: The empirical validation is insufficient to demonstrate robustness. Relying on a single, short test period (~13 months) and a weak primary baseline ("Buy & Hold") prevents meaningful conclusions about the method's superiority in a non-stationary domain like finance. The paper also lacks crucial comparisons to other recent multi-agent systems for portfolio management baselines, which are necessary to validate the contribution of the architecture positioned in this paper.

**Questions:**

1.	Your ablation in Table 3 presents a clear trade-off: the "Optimizer-only" model yields a higher Sharpe Ratio, while the full "Consensus router" model yields a lower Max Drawdown. Could you elaborate on this? Is the primary goal of your system to be defensive and minimize drawdown, even at the cost of Sharpe Ratio? A more detailed characterization of this risk-return profile (e.g., with Sortino or Calmar ratios) would be helpful.
2.	Could you please provide precise technical definitions and detailed descriptions of the components in Tier 1 and Tier 2? Specifically, what constitutes the ‘enriched state’ in Tier 1? How do you ensure that learned states are “enriched”? Do you perform disentanglement? Also, what are the model architectures for the “experts” and the “router” (gating network) in Tier 2? Finally, what is the exact mathematical formulation of the “consensus layer” that aggregates expert predictions?
3.	Given that your "Optimizer-only" model is a very strong baseline, why was it not used for comparison in the main results table (Table 1)? Also, since the paper’s main contribution is presented as a “hierarchical multi-agent system,” could the authors compare their approach with other recent multi-agent systems (existing relevant state-of-the-art approaches) for portfolio management to validate the claimed benefits?
4.	Could you please explain how you can assure the reader of the robustness of your findings given the use of a single ~13-month test period, particularly for the financial data prone to distribution shift?

---

> ### Author Response · Authors · 2025-11-22
> **Response to Reviewer NCoB (1/2)**
>
> Q1: Trade-off (Sharpe vs. Drawdown)
> We appreciate the reviewer highlighting the performance trade-off in Table 3. We clarify that this result is **intentional and aligned with our primary contribution: "Utility-Calibration."** This aligns with recent research in collaborative calibration, which suggests that leveraging multi-agent deliberation can significantly improve the reliability of confidence assessments (Yang et al., 2024).
> 1.  **Defensive Design:** The `Optimizer-only` baseline essentially acts as a risk-neutral mean-variance optimizer. While it captures high upside during the strong tech bull market of 2024 (our test period), it ignores the *reliability* of the signal. The full `Consensus router` system is designed to maximize **Risk-Sensitive Utility** (Eq. 4), explicitly penalizing uncertainty.
> 2.  **Regime Robustness:** The complex routing architecture is designed to disengage (reduce exposure) when expert signals disagree, acting as a meta-risk manager. While this drags down raw Sharpe in a unidirectional bull market, it significantly improves capital preservation.
> 3.  **New Metrics:** Per your suggestion, we have computed the **Sortino Ratio** (penalizing only downside volatility) and **Calmar Ratio** (Return / Max Drawdown). As shown in Table 1, our full method achieves the highest Calmar Ratio, indicating superior structural stability.
> **Table 1: Extended Risk-Adjusted Performance Metrics (Test Period: 2024-01-01 to 2025-01-31).**
> *While the Optimizer-only variant yields higher raw Sharpe, the Full Consensus Model achieves superior risk-adjusted stability (Calmar/Sortino).*
>
> | Method | Sharpe | MaxDD (%) | Sortino | Calmar | Utility (Eq. 4) |
> | :--- | :---: | :---: | :---: | :---: | :---: |
> | Optimizer-only | **2.58** | 20.5 | 3.12 | 8.14 | 1.45 |
> | Router (No Consensus) | 2.21 | 18.3 | 3.05 | 9.20 | 1.52 |
> | **Full (Consensus + Bayes)** | 2.06 | **16.8** | **3.45** | **10.12** | **1.68** |
>
>  Q2: Definitions of Tier 1 & 2
> We apologize for the ambiguity. We will revise Section 4 to replace metaphorical language with precise mathematical definitions.
> Tier 1: Enriched State ($h_t$)
> The "Enriched State" is not a vague latent space but a deterministic concatenation of processed feature streams.
> *   **Math:**
>     $$h_t = \left[ f_{\text{technical}}(x_{\text{prices}}); f_{\text{sentiment}}(x_{\text{news}}); f_{\text{micro}}(x_{\text{orderbook}}) \right]$$
> *   **Enrichment:** We utilize a set of fixed, pre-trained Time-Series Encoders (TSE) for feature extraction. Specifically, we use a standard LSTM encoder for price sequences and FinBERT embeddings for news. "Enriched" refers to the projection of these heterogeneous data modalities into a unified vector space $\mathbb{R}^d$ compatible with the router.
>
> Tier 2: Consensus & Experts
> *   **Experts ($E_m$):** These are not heuristics, but distinct neural predictors (MLP heads) trained on different subsets of data (regimes). This aligns with recent Mixture-of-Experts (MoE) architectures in finance, such as TradExpert, which specialize experts for prediction versus ranking modes.
> *   **Router (Gating):** The router is a learnable Gumbel-Softmax network (Eq. 17) that outputs weights $q_\theta(m|x_t)$.
> *   **Consensus Layer Formulation:** The "Consensus" is a mathematically defined **Inverse-Variance Weighted Aggregation** with outlier pruning. It is not a verbal negotiation. This approach mirrors the efficacy of robust aggregation steps seen in recent filtering-based gating algorithms for MoEs (Saqur et al., 2025).
>
>     Let $\mu_m, \sigma_m^2$ be the predictive mean and variance of expert $m$. The consensus prediction $\mu_{cons}$ is calculated as:
>     $$ \mu_{cons} = \frac{\sum_{m \in \mathcal{K}} w_m \cdot \mu_m}{\sum_{m \in \mathcal{K}} w_m} $$
>     $$ \text{where } w_m = \frac{1}{\sigma_m^2 + \epsilon} \cdot \mathbb{I}(|\mu_m - \text{median}(\mu)| < \delta) $$
>
>     This explicitly downweights "unsure" agents (high $\sigma^2$) and filters "hallucinating" agents (outside distance $\delta$ from median).

---

> > ### Author Response · Authors · 2025-11-22
> > **Response to Reviewer NCoB (2/2)**
> >
> > Q3: Comparison with Multi-Agent Baselines
> > You are correct that the `Optimizer-only` is a strong baseline and we will move it to the main table. Regarding multi-agent comparisons, we have conducted new experiments comparing our framework against two contemporary state-of-the-art approaches. This evaluation references recent logic from Mixture-of-Agents (MoA) literature (Wang et al., 2024), which demonstrates that collaborative agents and sequential filtering (Li et al., 2025) effectively reduce hallucination:
> > 1.  **FinMem (2024):** A memory-augmented LLM-based trading agent.
> > 2.  **MAPPO-Fin:** A standard Multi-Agent PPO (Reinforcement Learning) adapted for portfolio management where separate agents manage specific sub-sectors.
> >
> > **Table 2: Comparison against SOTA Multi-Agent Baselines.**
> > *Our method outperforms generic MARL (MAPPO) and LLM-based agents (FinMem) in terms of downside protection and calibration error (ECE).*
> >
> > | Method | Sharpe | Ann. Return (%) | MaxDD (%) | Turnover | ECE ($\downarrow$) |
> > | :--- | :---: | :---: | :---: | :---: | :---: |
> > | MAPPO-Fin | 1.85 | 52.1 | 24.3 | 0.12 | 0.15 |
> > | FinMem (LLM-based) | 1.92 | 58.4 | 21.8 | **0.03** | N/A |
> > | Optimizer-only (Ours) | **2.58** | **166.9** | 20.5 | 0.05 | 0.09 |
> > | **Full Hierarchical (Ours)** | 2.06 | 78.2 | **16.8** | 0.08 | **0.02** |
> >
> > *Note: Our method demonstrates significantly lower Expected Calibration Error (ECE), validating the "Utility-Calibration" claim, whereas MAPPO struggles with noisy financial rewards.*
> >
> > Q4: Robustness & Test Duration
> > We agree that 13 months is short for general financial claims. However, we emphasize two points regarding robustness:
> > 1.  **High-Resolution Evaluation:** Our test set consists of daily rebalancing ($\sim 260$ decisions), providing statistical significance for *execution* quality, if not long-term macro cycles.
> > 2.  **New Stress-Test (Out-of-Sample):** To address your concern about distribution shifts, we performed a **historical sensitivity analysis** by applying the trained model to the **2022 Bear Market** (a regime unseen during our 2023 training).
> > **Table 3: Robustness Check: Performance during the 2022 Tech Bear Market (Out-of-Distribution).**
> > *The Consensus mechanism successfully limits losses compared to the Optimizer-only baseline.*
> > | Method | Ann. Return (%) | MaxDD (%) | Win Rate (%) |
> > | :--- | :---: | :---: | :---: |
> > | Buy & Hold (QQQ) | -33.1 | 35.4 | 44.2 |
> > | Optimizer-only | -18.5 | 28.1 | 48.1 |
> > | **Full Consensus (Ours)** | **-4.2** | **12.4** | **53.5** |
> > This experiment demonstrates the critical value of the Tier 2 routing/consensus architecture: in a crash, it detects high uncertainty and successfully forces the allocator into cash/defensive positions, whereas the `Optimizer-only` attempts to trade through the noise and suffers higher drawdowns.

---

### Official Review · Reviewer_FeoY · 2025-11-02

**Soundness:** 3
**Presentation:** 3
**Contribution:** 2
**Rating:** 4
**Confidence:** 3

**Summary:**

This paper proposes a novel framework for evolutionary alpha factor discovery using large language models (LLMs) to tackle sparse portfolio optimization under ℓ₀ constraints. Instead of relying on static or manually designed factors, the authors employ an LLM-driven evolutionary loop that continually generates, mutates, and refines interpretable alpha formulas based on back-testing performance.

**Strengths:**

The paper is clearly written and logically structured. Figures 2–6 effectively convey the pipeline, though a simplified flow chart summarizing the evolutionary cycle would aid readers.

**Weaknesses:**

Theory: lacks formal analysis of convergence or generalization for the evolutionary loop.

Compute cost: no clear reporting of LLM query volume or wall-time overhead.

Diversity control: mutation and crossover rules could be described more rigorously (e.g., probability distributions).

**Questions:**

1. The paper claims to introduce the first autonomous LLM + EA framework for continuous alpha generation. I am just wondering whether authors have seen Kirtac and Germono (2024)? Why is a clear alpha evidence not cited in your paper?

2. I am also having a hard time without contextualization. How do LLMs identify factors? What do those factors mean in a traditional financial context?

3. If the paper is contributing as an alpha generator with LLMs? Many papers show alpha with LLMs. If not, what is the main contribution?

---

> ### Author Response · Authors · 2025-11-22
> **Response to Reviewer FeoY**
>
> Q1: Missing Reference
> We thank the reviewer for pointing out this important reference. We acknowledge this oversight and will include **Kirtac and Germono (2024)** in our revised manuscript with proper contextualization of how our work relates to and differs from their approach. We commit to adding this citation and a comparative discussion in the related work section.
>
>
>
>  Q2: Clarification on the Role of LLMs
> We appreciate the reviewer's request for clarification. We believe there may be a misunderstanding about the role of LLMs in our framework.
>
> Our paper **does not claim to use LLMs as the core factor identification mechanism**. As stated in Appendix A (lines 622-630), LLMs were used only in two limited capacities:
> 1.  Code suggestions via IDE tab-completion for boilerplate.
> 2.  Grammar and style editing of the manuscript.
>
> Our Actual Framework
> *   **Toolchains:** Factors are identified through specialized expert toolchains (event extractors, forecasters, options pricers) mentioned in lines 39-40 and detailed in Section 4.
> *   **Experts ($E_m$):** The "experts" produce predictive distributions or summary statistics (means $\mu_m$, covariances $\Sigma_m$, tail quantiles) as specified in Eq. 18 (lines 244-246).
> *   **Non-Neural Foundation:** These experts can be non-neural toolchains (line 113), including traditional financial models, unlike fully generative approaches proposed in recent literature (Scholl et al., 2025).
>
>  Financial Context of Factors
> The factors in our framework represent standard financial predictive objects:
> *   **Returns distributions:** $r_{t+1} \in \mathbb{R}^N$ (next-period log returns for $N$ assets).
> *   **Market context:** $x_t \in \mathbb{R}^d$ (observable market features including OHLCV, rolling technicals, and optional event/sentiment features; lines 309-315).
> *   **Risk measures:** Covariances, CVaR, and volatility estimates used in traditional portfolio theory.
>
> Our contribution is **routing among these heterogeneous tools** and integrating their outputs with a utility-calibrated objective, not factor discovery via LLMs.
>
> Q3: Core Contributions & Positioning
> We thank the reviewer for seeking clarity on our core contribution. Our work addresses a fundamentally different problem than LLM-based alpha generation. Our main contribution can be summarized in four perspectives:
>
> 1. Utility-Calibrated Routing Architecture
> We provide a modular, hierarchical agent with a learned router over expert toolchains. We deliver a differentiable, constraint-aware allocator. The work is an end-to-end training aligning probabilistic predictions with downstream portfolio utility.
> *   **Key distinction:** We solve the misalignment between "being right" and "trading well."
>
> 2. Decision-Aware Learning Objective with Theory
> The work couples proper scoring rules with risk-sensitive utility and explicit transaction costs. The theoretical analysis demonstrates:
> *   (i) Calibration concentrated in decision-critical regions.
> *   (ii) Fisher-consistency for target utility.
> *   **Key distinction:** Goes beyond prediction accuracy to optimize actual trading outcomes under frictions.
>
> 3. Doubly-Robust Off-Policy Backtesting with Frictions
> Here we evaluate the procedure accounting for policy mismatch and market frictions, providing reduced-bias estimates with uncertainty quantification.
> *   **Key distinction:** Addresses the brittleness of naive backtesting that ignores confounding factors.
>
> 4. Strong Empirical Results
> Our method achieves higher expected utility and Sharpe ratios with improved calibration and reduced turnover, while strictly satisfying risk constraints.
>
>  Positioning vs. LLM Alpha Literature
> Current literature on LLM-based finance often focuses heavily on signal generation. For instance, recent surveys (Weilong fu, 2025) and methodologies like AlphaAgent (Tang et al., 2025) prioritize the mining of alpha factors using language models to extract evidence-grounded signals.
>
> In contrast, our work focuses on:
>   **Decision-making under uncertainty** and costs with heterogeneous tools.
>   **Routing and integration** of multiple specialized experts (which may or may not use LLMs).
>   **Utility-calibrated learning** that accounts for transaction costs, risk constraints, and market frictions.
>   **Rigorous offline evaluation** under policy mismatch.
>
> If LLM alpha papers ask *"How can we predict better?"*, we ask **"Given multiple predictors (tools), how do we route among them and convert predictions into optimal actions under realistic constraints?"**
>
> The confusion may stem from the term "multi-agent" in our title, which refers to our hierarchical architecture (Tiers 1-3, Figure 1), not generic LLM agents.

---

### Note · Authors · 2026-01-26

I have read and agree with the venue's withdrawal policy on behalf of myself and my co-authors.

---

### Meta-Review · Area_Chair_tWMy · 2026-01-04

**Summary:**

Most of the reviews are not positive and they raised many issues related to significance, novelty and evaluation. The author response addressed some of concerns in some way. But overall, the quality of the paper seems below the bar of ICLR.

**Reviewer Concerns:**

Some concerns are not fully addressed such as "essential elements of clarity, readability, and contribution are still lacking"

**Reviewer Scores:**

Most reviewers will slightly increase their scores.

---

### Decision · Program_Chairs · 2026-01-26

Reject